# Identification of KFB Family in Moso Bamboo Reveals the Potential Function of PeKFB9 Involved in Stress Response and Lignin Polymerization

**DOI:** 10.3390/ijms232012568

**Published:** 2022-10-19

**Authors:** Kebin Yang, Ziyang Li, Chenglei Zhu, Yan Liu, Huayu Sun, Xueping Li, Zhimin Gao

**Affiliations:** 1Institute of Gene Science and Industrialization for Bamboo and Rattan Resources, International Center for Bamboo and Rattan, Beijing 100102, China; 2Key Laboratory of National Forestry and Grassland Administration/Beijing for Bamboo & Rattan Science and Technology, Beijing 100102, China

**Keywords:** *Phyllostachys edulis*, KFB, ubiquitination, abiotic stress, peroxidase activity, lignin polymerization

## Abstract

The Kelch repeat F-box (KFB) protein is an important E3 ubiquitin ligase that has been demonstrated to perform an important post-translational regulatory role in plants by mediating multiple biological processes. Despite their importance, KFBs have not yet been identified and characterized in bamboo. In this study, 19 PeKFBs were identified with F-box and Kelch domains; genes encoding these PeKFBs were unevenly distributed across 12 chromosomes of moso bamboo. Phylogenetic analysis indicated that the PeKFBs were divided into eight subclades based on similar gene structures and highly conserved motifs. A tissue-specific gene expression analysis showed that the *PeKFB*s were differentially expressed in various tissues of moso bamboo. All the promoters of the *PeKFB*s contained stress-related *cis*-elements, which was supported by the differentially expression of *PeKFB*s of moso bamboo under drought and cold stresses. Sixteen proteins were screened from the moso bamboo shoots’ cDNA library using PeKFB9 as a bait through a yeast two-hybrid (Y2H) assay. Moreover, PeKFB9 physically interacted with PeSKP1-like-1 and PePRX72-1, which mediated the activity of peroxidase in proteolytic turnover. Taken together, these findings improved our understanding of PeKFBs, especially in response to stresses, and laid a foundation for revealing the molecular mechanism of PeKFB9 in regulating lignin polymerization by degrading peroxidase.

## 1. Introduction

Post-translational modification (PTM), which plays a central role in the maintenance of protein stability and the modulation of protein activity, affects many aspects of plant development and growth [1,2]. Various types of protein modification have been reported, such as ubiquitination, acetylation, methylation, glycosylation, myristoylation, and phosphorylation [3]. Among them, ubiquitination is one of the main post-translational regulation mechanisms in eukaryotes; it distinguishes itself from others in that most of the ubiquitinated proteins are targeted to the 26S proteasome for degradation. Three sequential ubiquitin enzymes are involved in mediating ubiquitination: the ubiquitin-activating enzyme (E1); the ubiquitin-conjugating enzyme (E2); and the ubiquitin ligase (E3), which catalyzes the formation of an isopeptide linkage between the activated ubiquitin and the lysine residue of the substrate protein. Finally, these isopeptides target the 26S proteasome, where the target proteins are degraded; meanwhile, the ubiquitin monomers are reclaimed by the action of de-ubiquitination enzymes [4].

The specificity of protein ubiquitination is mainly determined by E3, and due to this, E3 provides recognition and binding specificity to the substrate in a temporally and spatially regulated manner [5,6]. The E3 superfamily is classified into single subunit proteins and cullin-based multi-subunit protein complexes, both of which can mediate ubiquitination; the multi-subunit protein complexes include SCF, CUL3-BTB, and CUL4-DDB [7]. F-box proteins form a ubiquitin ligase complex together with the suppressor of kinetochore protein 1 (SKP1), Cullin 1 (CUL1) and Ring-Box 1 (RBX1), where it plays the critical role of recruiting substrates to the ubiquitin–proteasome system [8]. F-box proteins were divided into different subfamilies based on classification criteria of variable protein–protein interaction motifs at the C termini, such as leucine-rich repeats (LRR), WD-40, and Kelch [9,10]. The Kelch repeat F-box (KFB) protein is one of these subfamilies; it contains C-terminal Kelch repeats but excludes the F-box motif in N-termini [11].

In plants, the KFB subfamily includes members with one to five Kelch repeats consisting of 44–56 residues [12]. To date, 46, 103, 31, and 36 KFBs have been described in *Oryza sativa*, *Arabidopsis thaliana*, *Triticum aestivum*, and *Salvia miltiorrhiza* respectively [11,12,13]. Several KFBs have been characterized and shown to have functions in regulating many important biological processes such as seed germination [14], hypocotyl elongation [15,16], panicle growth [17], leaf senescence [18], and other developmental processes [19,20]. Importantly, KFBs have three very typical functions. Firstly, KFBs are involved in regulating photoperiod flowering. FLAVIN-BINDING KELCH REPEAT F-BOX 1 (FKF1) is one of the most intensively researched FBKs, the expression of which is controlled by the circadian clock [21]. These proteins play important roles in modulating photoperiodic flowering [22,23,24] and the mechanisms of regulating proteins at both the transcriptional and post-translational levels in promoting flowering [25,26,27,28]. Secondly, KFBs are the result of the biosynthesis of secondary metabolites such as cuticular wax [29], phenols [30], flavonoids [31], and glucosinolate [32]. Lastly, several KFBs have been identified as key regulators in the phenylpropanoid biosynthesis pathway by means of physically interacting with enzymes of the phenylpropanoid pathway isoforms such as PAL and CCR to selectively mediate enzyme ubiquitination and proteolytic degradation via the 26S proteasome pathway [33,34,35]. Interestingly, these secondary metabolites play important roles in tolerance of stresses, suggesting KFB proteins have many regulatory roles in response to biotic and abiotic stresses.

To date, it has been challenging to face the depletion of forest resources and the increase in the protection of existing resources, so there is a need to develop and utilize nonwoody biomass [36]. In this context, bamboo, which offers a short growth cycle and equal physical and mechanical properties such as an excellent mechanical and tensile strength, has been widely used as an essential alternative to timber in mitigating its shortage [37]. The better physical and mechanical properties of bamboo are highly dependent on lignification. Studies in the field of the molecular mechanisms and genetic manipulations of lignification at the transcriptional level have been widely carried out and included lignin synthase genes [38,39], transcription factors [40,41], and miRNAs [42]; a reliable miRNA-mediated “MYB-*PeLAC20*” module for lignin monomer polymerization in bamboo was proposed in [42]. However, the underlying post-translational regulatory mechanisms of lignification in bamboo remain unclear. As the important E3 ubiquitin ligases, KFBs also play an important role in lignification and lignin biosynthesis. So, we performed identification and characterization analyses of the KFB gene family in moso bamboo. Detailed analyses including phylogenic distribution, structural features, chromosome location, and selection pressure for duplicated gene pairs were conducted. Moreover, the tissue- and stage-specific expression patterns of KFB genes in moso bamboo as well as the expression changes in leaves under drought and cold stresses were analyzed. Finally, to gain insights into the molecular mechanism of PeKFB9 involved in lignin polymerization, we systematically explored the potential protein–protein interactions of PeKFB9 with PeSKP1-like-1 and PePRX72-1, which laid a foundation for the future investigation of the potential functions of PeKFBs in bamboo.

## 2. Results

### 2.1. Identification and Characterization of KFB Genes in Moso Bamboo

Through BLASTP with the queries of Arabidopsis and rice KFBs in the bamboo database, we looked for conserved F-box domains and Kelch domains by Pfam and SMART in all the candidates and excluded any gene that encoded truncated KFB proteins. As a result, a total of 19 Kelch repeat F-box (KFB) proteins were obtained in moso bamboo (named as PeKFB1-PeKFB19) and used for further analysis. The characteristics of the proteins encoded by PeKFBs, including the numbers of amino acids and Kelch motifs, their molecular weight, and isoelectric point, were systematically analyzed and are listed in Appendix A. The amino acid lengths of PeKFBs ranged from 345 aa (PeKFB14) to 634 aa (PeKFB1) with molecular weights between 38.91 kDa and 68.29 kDa. We observed that PeKFBs had a variable theoretical isoelectric point with a high of 9.78 (PeKFB6) and a low of 4.91 (PeKFB14). All the PeKFBs were localized in the nucleus; PeKFB13 was also localized in the chloroplast. The conserved motif analysis showed that eight PeKFBs only had a single Kelch motif, four PeKFBs had two Kelch motifs, one had three Kelch motifs, and the remaining five had four Kelch motifs.

### 2.2. Phylogenetic Analysis of KFBs

To gain clear insight into the evolution of the PeKFBs, a phylogenetic analysis was performed using the protein sequences of KFBs in moso bamboo, Arabidopsis, and rice. The results showed that all the KFBs from different plant species could be clearly demarcated into eight major subclades and that the members of each subclade were different (Figure 1). Within the separated clades, the PeKFBs were clustered close to the KFBs of rice and distant from those of Arabidopsis. Notably, Subclades C and F only contained KFBs of Arabidopsis and rice with the absence of PeKFBs while the other subclades comprised the KFBs of three species. Subclade G comprised the maximum number of PeKFBs (6) followed by Subclade D (5), Subclade A (3), Subclades E and H (2), and Subclade A (1). Interestingly, At1g15670 (AtKFB01), At1g80440 (AtKFB20), and At3g59940 (AtKFB50), which are associated with lignin biosynthesis [33], were clustered in Subclade A, in which three PeKFBs (PeKFB7, PeKFB9, and PeKFB17) were clustered close to them.

### 2.3. Structural Features of PeKFBs

The structural features of all the identified PeKFBs were examined to gain insights into their evolution. It is well known that gene structural diversity might be an explanation for the evolution of multigene families. According to the predicted structures, 11 of the 19 *PeKFB*s had no introns while the rest of the *PeKFB*s only contained one intron (Figure 2a). Structurally, the closely linked *PeKFB*s were more analogous and generally showed the same exon/intron pattern in terms of intron number and position or exon length. *PeKFB6* and *PeKFB18* were the most representative genes and showed very similar exon–intron structures.

Furthermore, a conserved-motifs analysis showed that a total of 10 motifs were found in the PeKFBs (Figure 2b). Fourteen PeKFBs shared motif 5 and motif 6 while the remaining five PeKFBs all contained motif 1. Motif 1 and motif 5 constituted the most highly conserved part of the F-box domain and F-box-like domain. Interestingly, some motifs were detected only in specific KFBs; for example, motif 2 was only found in PeKFB3. Concretely, each Kelch motif was featured with three conserved amino acid residues: two adjacent glycines (G) and one nonadjacent aromatic amino acid (W). However, the amino acid residues of the Kelch motif domain at the C-termini of KFBs lacked strictly conserved sequences and only a few amino acid residues were relatively invariant (Appendix A). The variance in the motif patterns might be related to diverse functions of PeKFBs within the same subclade.

### 2.4. PeKFBs Distributed in Chromosomes and Syntenic Analyses

The coordinates of 19 *PeKFB*s were extracted from the moso bamboo GFF file to analyze their localization on the moso bamboo genome. The distribution of the *PeKFB*s on the pseudochromosome is shown in Figure 3a, which reveals that the *PeKFB*s were distributed on 12 chromosomes with different densities. Chromosome 20 contained the highest number of *PeKFB*s (four), followed by chromosome 17 with three *PeKFB*s and chromosome 5 with two *PeKFB*s, while the other chromosomes anchored only one. To understand the duplication events of *KFB*s in moso bamboo and rice, a synteny analysis was performed using MCscanX and the Advanced Circos software in TBtools. The results demonstrated that nine segmental duplication pairs were found within *PeKFB*s and 23 segmental duplication pairs were detected between *PeKFB*s and *OsKFB*s (Figure 3a). 

To better recognize the evolutionary selection pressure for *KFB*s in moso bamboo and rice, we identified all 32 pairs of duplicated *KFB*s. We calculated the *Ka* and *Ks* values and *Ka*/*Ks* ratios of duplicated KFB gene pairs (Appendix A). The *Ka*/*Ks* ratios of all duplicated gene pairs were less than 1.0 (Figure 3b,c), suggesting that the purification selection existed in *PeKFB*s during moso bamboo’s evolution.

### 2.5. Responsive and Cis-Elements in the Promoters of PeKFBs

The 2000 bp sequences upstream of the *PeKFB*s’ encoding region were predicted using the PlantCARE database. The results revealed that there were many responsive and cis-elements associated with hormones and abiotic stresses (Figure 4). A large number of stress-responsive elements including the anaerobic induction element (ARE and GC-motif), low-temperature-responsive element (LTR), and MYB binding site (MBS) were found to be abundant in the promoters of the *PeKFB*s. Most promoters of the PeKFBs contained MYB and MYC binding elements related to drought and cold (Figure 4). These results suggested that these *PeKFB*s may participate in the response to various stresses. In addition, some promoters of *PeKFB*s contained a cis-element involved in MeJA responsiveness and abscisic acid responsiveness. In particular, ABRE, the CGTCA motif, and the TGACG motif were enriched in almost every promoter of the *PeKFB*s. In addition, many types of cis-elements involved in morphogenesis and organogenesis were identified, such as an element involved in differentiation of the palisade mesophyll cells and an element involved in seed-specific regulation. Moreover, the promoters of the *PeKFB*s also contained vast light response elements such as the G-Box, GATA and GT1 motifs (Appendix A). 

### 2.6. Expression Patterns of PeKFBs in Different Tissues of Moso Bamboo

The gene expression patterns could provide information for potential gene functions. The recent transcriptome data generated from different tissue samples was used to investigate the expression patterns of the *PeKFB*s (Figure 5). Accordingly, 19 *PeKFB*s were classified into five separate groups (Group I~Group V) in the heat map according to the hierarchical clustering of expression patterns; the genes in Group I–Group III generally showed a higher expression abundance than those in other groups. Group I consisted of four *PeKFB*s that had a higher expression level in young tissues. For example, PeKFB9 showed a high expression in 0.2 m shoots, 0.1 cm roots, and buds; however, its expression pattern was not significant in Group II. Group III included five *PeKFB*s (*PeKFB2*/*10*/*14*/*16*/*17*) that had high transcript levels in all stages and tissues. A relatively low transcript expression level of these genes was observed in older roots, shoots, and bud tissues; a high expression abundance of them was detected in 0.2 m shoots and rhizome. Similarly, they showed higher expression in the top and middle than those in the low parts of single internodes (Appendix A). These results also indicated that the *PeKFB*s in Groups I-III might be directly or indirectly involved in the rapid growth of bamboo shoots. Group IV included only two members (*PeKFB6* and *PeKFB7*) that were almost undetected in all tissues. The *PeKFB*s in Group V showed obvious tissue-specific effects correlated with the different developmental stages and the tissues. For instance, *PeKFB13* showed a relatively dominant expression level in blades but a relatively lower one in the other tissues.

### 2.7. Expression Patterns of PeKFBs in Leaves of Moso Bamboo under Drought and Cold Stresses

To further confirm whether the expressions of the PeKFBs were affected by stresses, except for 4 *PeKFB*s without suitable specific primers, 15 *PeKFB*s were carefully selected for RT-qPCR analysis according to their specific expression patterns in different tissues (Figure 5) and the elements in their promotors. Except for *PeKFB12*, the other 14 *PeKFB*s showed significantly upregulated expression patterns under cold stress compared with the control (Figure 6), which had three expression patterns: a continuously increasing trend, an increasing trend with a final decrease, and a trend of an initial decrease followed by a subsequent increase or slight stability with the treated time. The drought stress induced significant expression changes of 14 *PeKFB*s while *PeKFB5* showed no obviously significant expression changes (Figure 6). The expression of seven *PeKFB*s were significantly upregulated and then decreased, of which six reached the highest level at 6 h, whereas *PeKFB19* showed a peak at 3 h. *PeKFB8*/*15*/*16* was continuously upregulated during the drought treatment. The expression levels of *PeKFB9*/*10*/*12* were decreased significantly at 3 h and 6 h but were induced at 12 h under drought stress. 

### 2.8. Yeast Two-Hybrid Screening for the Proteins That Interacted with PeKFB9

PeKFB9 showed that it was highly homologous to AtKFB01, AtKFB20, and AtKFB50, which was related to phenylpropanoid biosynthesis, and had abundant transcription levels in different tissues of moso bamboo; therefore, PeKFB9 was selected for further analysis. The full-length cDNA sequence of PeKFB9 contained an ORF of 1092 bp encoding a putative 364 amino acid protein. A sequence analysis using the Pfam website indicated that PeKFB9 contained an F-box domain in the N-termini and a Kelch repeat domain near the C-terminus (Figure 7a). To identify the proteins that may interact with PeKFB9 while participating in the lignification of moso bamboo, a cDNA library of moso bamboo shoots was constructed and a yeast two-hybrid system was conducted to screen for the interactors to find the potential chaperones of the PeKFB9 protein using pGBKT7-PeKFB9 as bait. 

Among 16 positive clones, 2 clones were characterized by containing the same protein (encoded by PH02Gene22472) (Appendix A). The deduced protein of this gene belonged to the SKP1-like proteins, which are the core subunit of the SKP1/Cullin/F-box E3 ubiquitin ligase complex. However, the function of the identified protein here was still unknown, so the protein was named the PeSKP1-like-1. A Y2H assay was used to confirm the potential physical interaction between PeKFB9 and PeSKP1-like-1. The pGBKT7-PeKFB9 and pGADT7-PeSKP1-like-1 plasmids were co-transformed into yeast AH109 cells and the co-transformed yeast cells were grown on synthetic dropout medium (Figure 7d). Next, a truncated expression vector of F^PeKFB9 was constructed in which the predicted F-box domain of PeKFB9 was removed (Figure 7b). F^PeKFB9 showed an indiscernible interaction with other SKP1-like-1 in Y2H assays under the same conditions (Figure 7d). The results suggested that the N-terminal F-box motif of the KFB proteins is required to mediate the interaction of the KFB proteins with SKP1 to form a functional SCF-type E3 ligase.

### 2.9. PeKFB9 Is Involved in Regulating Lignin Polymerization by Degrading PePRX72-1

In addition, another candidate protein (PH02Gene23181) was identified by the Y2H screening that belonged to the peroxidase protein family and was homologous with AtPrx72, which had been shown to participate in lignin polymerization [43], so we named it PePRX72-1. In order to verify the function of PeKFB9 in moso bamboo, we first examined whether PeKFB9 physically interacted with PePRX72-1 via Y2H assays. The results showed that the yeast transformants harboring the expressed vectors of PeKFB9 and PePRX72-1 could effectively activate the expression of four independent reporter genes (lacZ, HIS3, ADE2, and MEL1), resulting in good growth of blue yeast colonies on an SD/-Ade/-His/-Leu/-Trp medium containing X-α-Gal (Figure 7d). Kelch repeat domains near the C-terminus were responsible for interacting selectively with the target proteins, therefore conferring specificity to this complex [44]. So, we constructed a truncated K^PeKFB9 in which we removed the predicted Kelch repeat domains of PeKFB9 (Figure 7c); the Y2H assay results showed that experimental group did not grow on the SD/-Ade/-His/-Leu/-Trp medium containing X-α-Gal (Figure 7d). 

Furthermore, we verified whether the interaction of PeKFB9 with the PePRX72-1 decreased peroxidase activity in order to evaluate the hypothesis that PeKFB9 mediated the degradation of the peroxidase. The assay of the transient expression of PePRX72-1 in tobacco leaves showed that co-expression of the PePRX72-1-FLAG fusion proteins with PeKFB9 decreased peroxidase activity more than 60% compared with the control (Figure 8a). To verify whether the reduction in peroxidase activity resulted from the degradation of the PePRX72-1 protein, we examined the stability of the PePRX72-1-FLAG fusions using Western blotting with an anti-FLAG antibody. The signal for PePRX72-1-FLAG fusions was weakened in the extracts from leaves co-expressing PePRX72-1 with the full-length PeKFB9 protein. However, the signal for PePRX72-1 was essentially unchanged when it was co-expressed with F^PeKFB9 (Figure 8b). In conclusion, PeKFB9 could inhibit the peroxidase activity by degrading PePRX72-1, resulting in the regulation of lignin polymerization.

## 3. Discussion

KFB proteins play crucial roles in the course of plant growth and in the response to biotic and abiotic stresses, as well as in phenylalanine biosynthesis [30,45]. Lignin biosynthesized from phenylalanine is deposited in mature tissues, which leads to lignification, which in turn affects the application value of timber and the texture, taste, and nutrition of fruits and vegetables during storage [42,46]. Lignin biosynthesis goes through two processes of phenylalanine biosynthesis and monomer polymerization. KFBs probably play a role in the lignification of bamboo and in response to environmental stress. Therefore, identification and characterization of KFBs in bamboo will be helpful in further research on bamboo growth. However, none of the KFBs were previously identified and reported in bamboo. In the present study, we conducted a systematic investigation of KFBs in moso bamboo and provided a basis for analyzing their functions in the future.

We identified 19 PeKFBs in the genome of moso bamboo, which was less than the numbers previously found in *Arabidopsis*, rice, potato, wheat, and chickpea [33,45,47,48]. A similar phenomenon occurred in the F-box family, suggesting that the KFB family in moso bamboo has experienced family contraction. The number and species-specific conserved motifs of each KFB family member were different to some extent but the conserved motifs and species-specific members in the same clade were roughly consistent (Figure 2). We found that three PeKFBs (PeKFB7, PeKFB9, and PeKFB17) in the same clade had the same gene structures and conserved motifs (one motif 5 and two motif 6), which suggested that they may have similar functions. The characterized domains (F-box domain and Kelch domain) of the PeKFBs had a lack of strictly conserved sequences, resulting in the differentiation and diversity of their functions (Appendix A); this was in agreement with previous studies [45,49]. In addition, the Kelch domain is responsible for selectively interacting with target proteins and its diversity suggests the recognition of different substrates such as PAL and S-ADENOSYL-L-METHIONINE SYNTHETASE1 [18]. In contrast with previous studies [45,50], KFBs including five Kelch domains had not been identified previously in moso bamboo, illustrating the contraction of the PeKFBs in their evolution with the corresponding functions also lost. These results of variation in the conserved motifs also suggested that the function of the PeKFBs was diversified during evolution.

The structural diversity and complexity of PeKFBs make their functions diverse as well, which was demonstrated in the diversity of the expression patterns. We further analyzed the expression patterns of the *PeKFB*s in different tissues and found a large variety among different *PeKFB*s (Figure 5), suggesting their diversified functions. The promoter region of a gene is related to its function; thus, the analysis of *cis*-elements assists in its functional characterization. Our results showed that the *PeKFB*s contained various *cis*-elements in their promoters, including *cis*-elements essential for light, phytohormone, and stress responses (such as drought and cold stresses) (Figure 4 and Appendix A), suggesting that these genes were involved in multiple biological processes, especially in response to stresses. The *PeKFB*s shared common drought- and cold-related *cis*-elements, so we performed RT-qPCR to verify their expression levels under drought and cold stresses. The results showed that *the PeKFB*s were induced or inhibited by these stresses, which was similar to the results found in other F-box protein genes [50,51]. Among the *cis*-elements, ABRE, as an important component in the ABA pathway, has been shown to be bound by transcription factors in response to ABA-mediated osmotic stress signals [52,53]. ABRE distributed in almost all the promoters of the *PeKFB*s (except for *PeKFB9*, *PeKFB12,* and *PeKFB17*). Therefore, it was consistent with the results of previous studies that these genes may be involved in the abiotic stress response and participate in the regulation of ABA signaling [54,55]. Meanwhile, only sporadic GA and IAA-related *cis*-elements were distributed in the promoters of a few *PeKFB*s, which was consistent with the results showing that the transcriptions of the *PeKFB*s were not responsive to these phytohormones (Appendix A), but did not agree with the results in tomato [45]. Interestingly, the homologous genes of At*KFB*s associated with phenylalanine biosynthesis (*PeKFB7*, *PeKFB9,* and *PeKFB17*) had the same structure but showed different expression patterns (Figure 2 and Figure 4), which was confirmed by the RT-qPCR results. These results suggested that the function of *PeKFB7* might be redundant to *PeKFB9* and *PeKFB17*, which would contradict the above assumption.

Lignification is regulated in transcriptional regulation, translational regulation, post-translational modification and proteolysis, and product inhibition. With regard to lignification in bamboo, most studies focused on transcriptional regulation [38,42]. To date, AtKFB01, AtKFB20, AtKFB39, and AtKFB50 have been identified as the interacting protein of AtPALs, which mediate PAL ubiquitination and subsequent degradation to negatively regulate plant phenylpropanoid biosynthesis [33]. PeKFB9 was phylogenetically related to AtKFB01, AtKFB20, AtKFB39, and AtKFB50, suggesting that PeKFB9 may share a similar function in moso bamboo. In this study, PePRX72-1 (PH02Gene23181) was identified by the Y2H assay using PeKFB9 as bait and was highly homologous to peroxidase 72 of *Arabidopsis,* which altered cell wall and phenylpropanoid metabolism. Class III peroxidases are glycoproteins with a major role in lignin formation and have been implicated in the cross-linking of lignin formation [56,57]. In rice, OsPRX38 facilitated lignin polymerization by oxidizing lignin monomers (monolignols) to promote lignin content [58]. Several PRXs in *Arabidopsis* were implicated in lignification, including AtPRX2, AtPRX4, AtPRX17, AtPRX25, AtPRX52, and AtPRX71, due to altered lignin content or composition in mutant plant lines [59,60,61,62], which was further supported by the assay of the transient expression of PePRX72-1 in tobacco leaves. In addition, PeKFB9 might have interacted with SKP1-like-1 to form a functional SCF-type E3 ligase (Figure 7), which promoted the ubiquitination and degradation of peroxidase. However, another subunit of the SCF-type E3 ligase (except for SKP1-like-1 and KFB9) was not identified, which is worthy of further experimental verification. 

## 4. Materials and Methods

### 4.1. Sequence Retrieval and Identification of Kelch Repeat F-Box (KFB) Proteins in Moso Bamboo

To obtain nonredundant KFB proteins in moso bamboo comprehensively, a BLASTP algorithm-based search provided by the BambooGDB database under the E-value cutoff of 1e-10 was first performed using the KFB protein sequences reported in Arabidopsis and rice [11]. The resulted sequences were then subjected to Pfam (https://pfam.xfam.org, accessed on 19 April 2022, European Molecular Biology Laboratory, Cambridgeshire, UK) to detect the presence and number of the F-box and Kelch domains. All KFB sequences were manually inspected and those that did not contain the conserved domains were discarded. Finally, the KFB proteins were named according to their BambooGDB assembly names in order.

The basic characteristics of the identified KFB proteins in moso bamboo were further analyzed, including the predicted proteins and the physicochemical parameters. The predicted molecular weights (MWs) and isoelectric points (pIs) of the KFB proteins were analyzed using ProtParam (). The subcellular localization was predicted by using Plant-mPLoc (http://www.csbio.sjtu.edu.cn/bioinf/plant-multi/# accessed on 19 April 2022).

### 4.2. Multiple Sequence Alignment and Phylogenetic Analysis of KFBs

To explore the evolutionary relationships among the KFBs in moso bamboo, MEGA 6.0 was used to perform multiple sequence alignments with the full length of KFB amino acid sequences and the maximum-likelihood phylogenetic unrooted trees were constructed using the FastTree program [63]. The KFB sequences of 103 AtKFBs and 39 OsKFBs were downloaded for the *Arabidopsis* genome from TAIR (The Arabidopsis Information Resource) release 10.0 (http://www.arabidopsis.org accessed on 19 April 2022) and the rice genome annotation database (http://rice.plantbiology.msu.edu/index.shtml, release 7.0 accessed on 19 April 2022) respectively, which were used for the construction of the phylogenetic trees.

### 4.3. Structural Features Analyses of KFB Family in Moso Bamboo

The gene structure of the KFB genes in moso bamboo was predicted using TBtools software [64]. The GFF3 file containing information about the intron–exon structure of the KFB genes was used as input. Multiple Expectation Maximizations for Motif Elicitation (MEME, http://meme-suite.org accessed on 19 April 2022, National Institutes of Health, Bethesda, Rockville, MD, USA) was adopted to identify the conserved motifs with the following parameters: the distribution of motif occurrences—zero or one per sequence; the maximum number of motifs—10; and the optimum width of each motif—between 6 and 300 residues [65].

### 4.4. Physical Localization and Gene Duplication Analysis of KFB Genes in Moso Bamboo

The chromosomal localization data of each KFB gene identified in moso bamboo was retrieved from the GFF file using TBtools. MCScanX software was used to identify the duplicated and syntenic KFB genes within the moso bamboo genome and those between moso bamboo and rice using the default settings [66]. The chromosomal distributions and syntenic relationships of the KFB genes were visualized using Circos software [67]. The syntenic analysis maps were constructed using the TBtools software to display the syntenic relationships of the orthologous KFB genes in moso bamboo and rice. The nonsynonymous (*Ka*) and synonymous substitution rates (*Ks*), as well as the *Ka*/*Ks* ratios of syntenic KFB gene pairs, were calculated using KaKs_Calculator software [68].

### 4.5. Putative Cis-Element Analysis in the Promoters of KEB Genes in Moso Bamboo

To obtain the promoter sequences of KFB genes in moso bamboo, we downloaded 2000 bp of the sequence upstream of each KFB gene coding region from the BamGDB. Cis-elements were analyzed using the database of PlantCARE (http://bioinformatics.psb.ugent.be/webtools/plantcare/html/ accessed on 19 April 2022). The number of elements in the promoters was shown in heat map made by using TBtools software.

### 4.6. Expression Analysis of KFB Genes in Moso Bamboo Based on RNA-Seq Data

To investigate the expression of KFB genes in the different tissues and those in shoots at different developmental stages of moso bamboo, the RNA-seq data generated from 26 samples of rhizomes, roots, shoots, leaves, sheaths, and buds were downloaded from the NCBI Short Read Archive (SRA) and used for further analysis. The accession numbers in SRA were SRX2408703~SRX2408728 and SRR13201212-SRR13201244, respectively. The RNA-seq data of moso bamboo leaves under NAA and GA_3_ treatments were also used for the expression profiles of the KFB genes. The expression levels were normalized by log_2_(FPKM+1) and loaded into TBtools software to generate heat maps. 

### 4.7. Real-Time PCR Analysis (RT-qPCR)

Sample collection: moso bamboo seeds gathered from the Guangxi Province of China were sown and cultivated in a greenhouse (25 ± 2 °C; 16 h/8 h light/dark cycle). After three months, the seedlings were used for the following treatments. Drought and cold stresses were simulated using a 20% PEG 6000 solution and 4 °C, respectively. The solution was poured into the cultured pots of moso bamboo seedlings and the leaves were harvested at 3 h, 6 h, and 12 h, respectively, and immediately frozen in liquid nitrogen and stored at −80 °C for RNA isolation. Meanwhile, the untreated seedlings were also collected and used as the control (0 h).

RNA extraction: the total RNA was extracted using a plant RNA extraction kit (Qiagen, Shanghai, China) according to the manufacturer’s instructions. The integrity of the total RNA was verified via agarose gel electrophoresis and the purity and concentration of the total RNA was determined via spectrophotometry (Nanodrop2000, Thermo, Waltham, MA, USA). The first strand cDNA was synthesized using a reverse-transcription system (Promega, Madison, WI, USA). For each 20 μL reaction, 1000 ng of total RNA was used; the synthesis was performed at 42 °C for 45 min. The final cDNA product was diluted 5-fold prior to use.

RT-qPCR: the specific primers for different KFB genes of moso bamboo were designed in Primer Premier 5.0 software and empirically adjusted for gene expression analysis (Appendix A). Additionally, all primers showing a clear specific melting peak according to a real-time melting curve analysis that was consistent with the results of agarose gel electrophoresis of specific PCR products were used for further analysis. The RT-qPCR was performed with a Roche Light Cycler 480 SYBR Green I Master kit on a qTOWER2.2 system (Analytik Jena, Jena, Germany). The PCR program involved 95 °C for 10 min followed by 40 cycles at 95 °C for 10 s and 60 °C for 10 s. The 10.0 μL reaction volume contained 5.0 μL of 2 × SYBR Green I Master Mix, 0.8 μL of cDNA, 0.1 μL of primer (10.0 mM each), and 4.0 μL of ddH_2_O. NTB was used as a reference gene [69] and the 2^−ΔΔCt^ method was used for the analysis and visualization of the generated PCR data [70].

### 4.8. Yeast Library Construction and Yeast Two-Hybrid (Y2H) Screening 

The moso bamboo cDNA library for Y2H experiments was constructed by OE BioTech (Shanghai, China) using the cDNA synthesized from the mRNAs of shoots at different developmental stages and cloned into the prey vector pGADT7. The full-length CDS of *PeKFB9* (PH02Gene23593.t1) was cloned into the pGBKT7 bait vector (Appendix A) and the fusion plasmids were further used for self-activation and Y2H screening of the pGADT7-based cDNA library. Y2H experiments were conducted using Matchmaker GAL4 two-hybrid systems; the putative PeKFB9-interacting clones were identified on an SD/-Ade/-His/-Leu/-Trp/X-α-Gal medium and further characterized and sequenced. To ensure the reliability of the library screening results, the interacted clones were cloned into the pGADT7 prey vector. Two recombinant vectors were co-transformed into the yeast strain AH109. The transformants were selected on SD/-Trp/-Leu and SD/-Trp/-Leu/-Ade/-His/X-α-Gal plates at 30 °C for five days to verify whether they were interacted. The co-transformants with pGBKT7-Lam and pGADT7-T were used as a negative control while the pGBKT7-53 and pGADT7-T combination were selected as a positive control. 

### 4.9. Assays for the Transient Expression of PeKFB9 and PePRX72-1 in Tobacco Leaves

The ORFs of *PeKFB9* and *PePRX72-1*, as well as the trunked *PeKFB9* (*F^PeKFB9* and *K^PeKFB9*), were cloned via PCR using the primers described in Appendix A. The PCR fragments were recombined into pCAMBIA1300-3×FLAG using the Infusion^®^ cloning method to generate PeKFB9-FLAG, F^PeKFB9-FLAG, and PePRX72-1-FLAG, respectively. The Agrobacterium strain GV3101 carrying the PePRX72-1 expression vector (OD_600_ = 0.6) was mixed with an equal volume of an Agrobacterium strain (OD_600_ = 0.6) carrying PeKFB9-FLAG, F^PeKFB9-FLAG, and empty pCAMBIA1300-3×FLAG, respectively; these were then used to infiltrate the tobacco leaves.

After three days of infiltration, the total soluble proteins were extracted with lysis buffer containing 20 mM of Tris-HCl (pH = 8.0), 150 mM of NaCl, 1 mM of EDTA, 10% glycerol, 0.2% Triton X-100, and 1 × protease inhibitor cocktail. The ratio of leaf fresh weight to extraction buffer was 1 g ml^−1^. The concentration of total protein was measured by using a Bradford protein assay kit (TaKaRa, Dalian, China). The proteins were examined using immunoblotting with an anti-FLAG monoclonal antibody.

In addition, harvested leaves (0.1 g) frozen at −80 °C were crushed with a prechilled mortar and pestle using an appropriate volume of prechilled extraction buffer. After uniform homogenization and centrifugation at 13,000 rpm for 15 min at 4 °C, the supernatant containing the crude enzymes was collected and used for the peroxidase activity analyses using the kit and following the manufacturer’s instructions (Suzhou Grace Biotechnology Co., Ltd., Suzhou, China).

### 4.10. Statistical Analysis

Analyses were performed with SPSS Statistics for Windows (Version 22.0. SPSS Inc., Chicago, IL, USA). All data were the average and standard deviation (SD) of three biological replicates. One-way analysis of variance was used to evaluate the statistical significance of differences among means using SPSS software. Different letters indicate respective significant differences at the level of *p* < 0.01.

## 5. Conclusions

In the present study, we conducted a comprehensive analysis of the KFB family in moso bamboo for first time. A total of 19 PeKFBs were identified and classified into eight subclades. The *PeKFB*s were mapped to 12 chromosomes unevenly; their exon-intron organization and the conserved motifs in their coding proteins were relatively similar to those in other monocot plants. The expression patterns of the *PeKFB*s in different tissues and organs at various developmental stages implied that they may be involved in the growth and development of moso bamboo. The *cis*-elements in the promoter regions of the *PeKFB*s and their expression patterns indicated that the *PeKFB*s might be involved in responses to drought and cold stresses. The results of the peroxidase activity and Western blotting analyses indicated that PeKFB9 was a post-translational regulator responsible for the turnover of peroxidase to negatively control the monomer polymerization. Taken together, the results of the present study provided an extensive resource for understanding the diversity of KFB genes that can be utilized for genetic manipulation in improvement of abiotic stress tolerance and wood properties in bamboo.

## Figures and Tables

**Figure 1 ijms-23-12568-f001:**
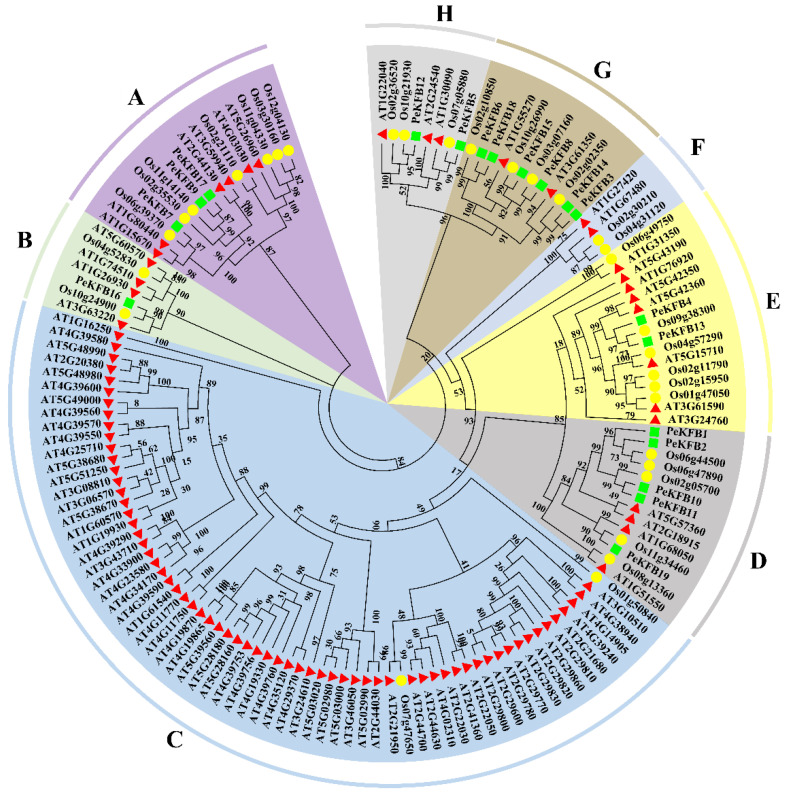
Phylogenetic analysis of KFB proteins in moso bamboo, *Arabidopsis,* and rice. Green squares, red triangles, and yellow circles represented moso bamboo, *Arabidopsis,* and rice, respectively. KFB proteins were categorized into eight subclades (**A**–**H**), which are highlighted in various colors.

**Figure 2 ijms-23-12568-f002:**
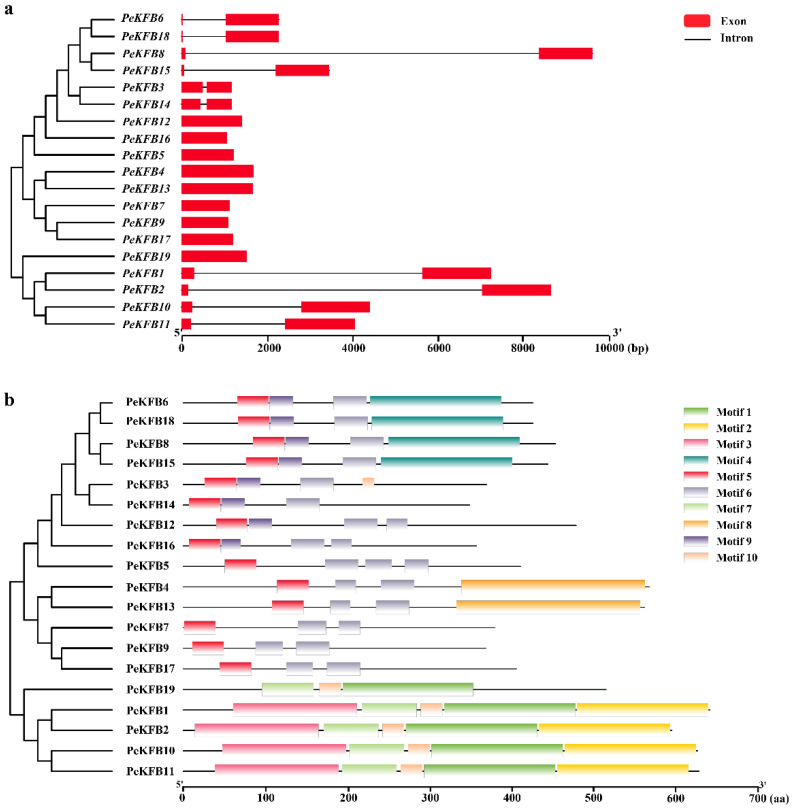
Structural features of KFB family members in moso bamboo. (**a**) Gene structures of *PeKFB*s. Red boxes: exons; lines between the boxes: introns. The scale bar at bottom demonstrates the length of exons and introns. (**b**) Conserved motifs of PeKFBs. Motifs 1–10 are presented in boxes with different colors. The motif information was obtained from the MEME website and visualized in TBtools.

**Figure 3 ijms-23-12568-f003:**
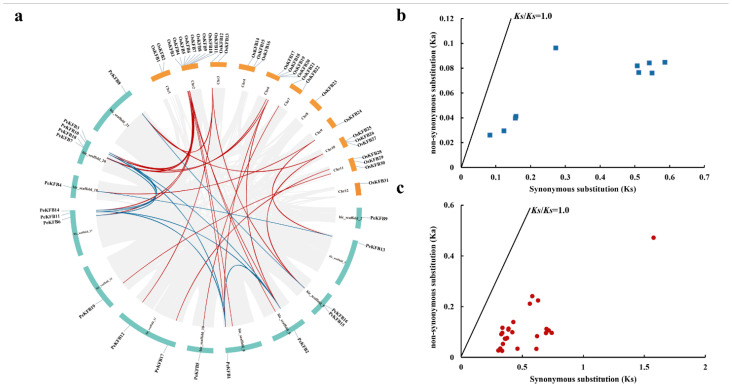
Synteny regions and *Ka*/*Ks* ratio analysis of *KFB*s. (**a**) Chromosome location and synteny regions of *PeKFB*s and *OsKFB*s. The duplicated genes of *PeKFB*s and *PeKFB*s/*OsKFB*s on different chromosomes are indicated with blue and red lines, respectively. (**b**) The *Ka*/*Ks* ratios of the gene pairs of *PeKFB*s. (**c**) The *Ka*/*Ks* ratios of the gene pairs of *PeKFB*s and *OsKFB*s.

**Figure 4 ijms-23-12568-f004:**
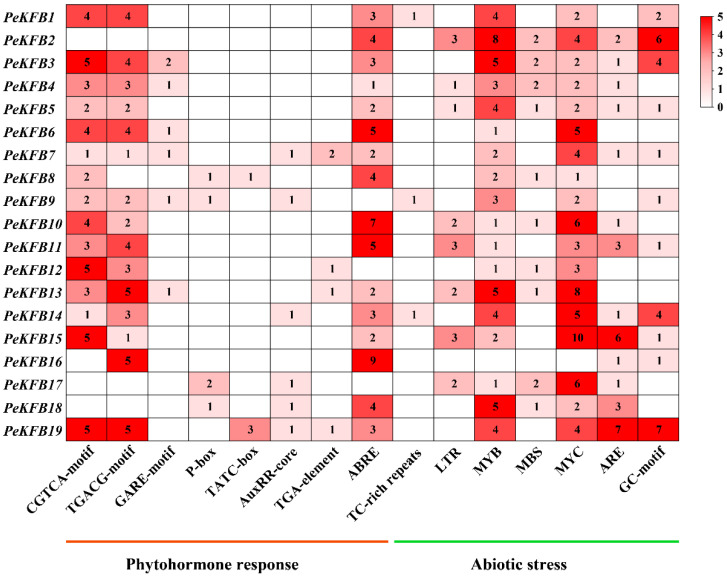
The analysis of conserved *cis*-elements associated with phytohormone response and abiotic stress in the promoter regions of *PeKFB*s. The 2000 bp promoter sequences were used to analyze nine specific phytohormone-related (auxin, gibberellin, MeJA, and abscisic acid) *cis*-elements and three abiotic stress-responsive (anaerobic induction, drought, and low-temperature) *cis*-elements. The right heatmap shows the number of *cis*-elements with higher numbers in dark red and lower numbers in light red.

**Figure 5 ijms-23-12568-f005:**
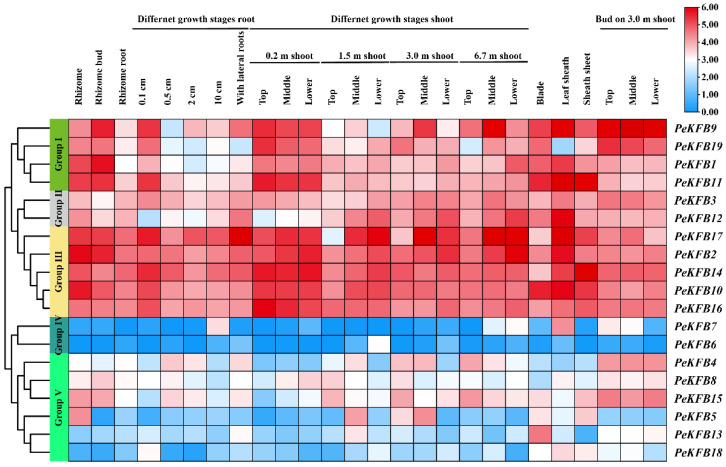
The expression patterns of 19 *PeKFB*s in different tissues based on RNA-seq data of moso bamboo. Hierarchical clustering heat maps were plotted based on FPKM values of 19 *PeKFB*s. Five groups are indicated by different colors on the left. The right heatmap shows gene expression with high levels in red and low levels in blue.

**Figure 6 ijms-23-12568-f006:**
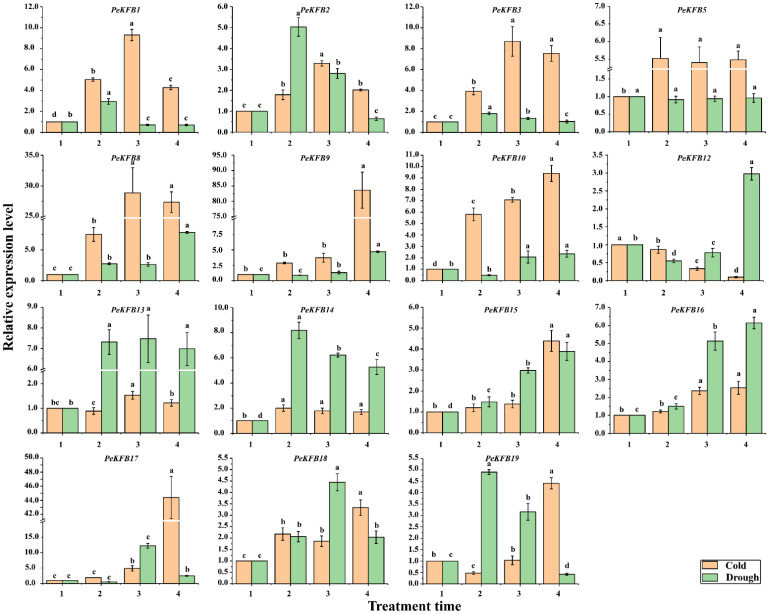
Expression patterns of *PeKFB*s in leaves of moso bamboo under low temperature and drought stresses. Different letters represent significant differences among different samples at *p* < 0.01. The average of three biological replicates along with the standard deviation (SD) are also given. The numbers 1, 2, 3, and 4 represent 0 h, 3 h, 6 h, and 12 h under stress treatments, respectively.

**Figure 7 ijms-23-12568-f007:**
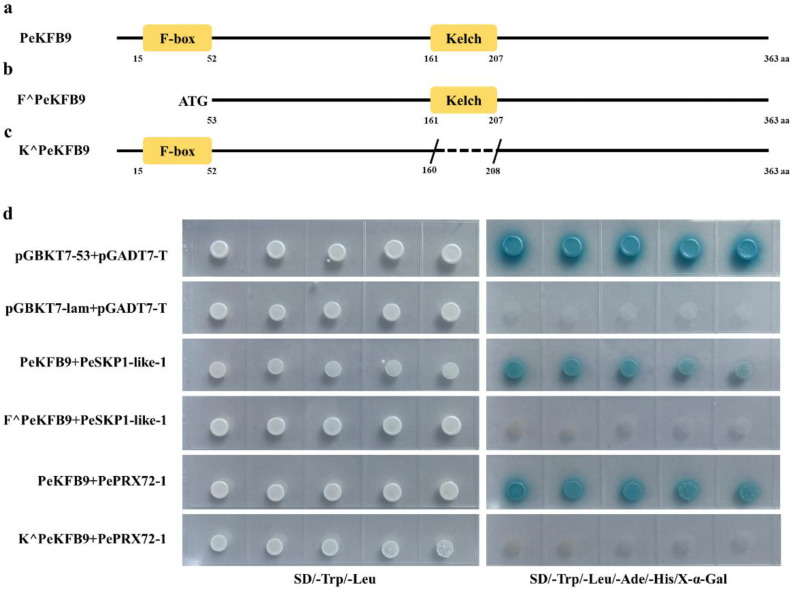
PeKFB9 physically interacted with PeSKP1-like-1 and PePRX72-1. (**a**–**c**) Schematic diagram showing the predicted F-box and Kelch repeat domains in PeKFB9, F^PeKFB9, and K^PeKFB9. (**d**) Yeasts carrying activation domain (AD) (pGADT7-PePRX72-1 and pGADT7-PeSKP1-like-1) and binding domain (BD) (pGBKT7-PeKFB9, pGBKT7-F^PeKFB9, and pGBKT7-K^PeKFB9) vectors grown on double dropout (-Leu/-Trp) and quadruple dropout (-Leu/-Trp/-His/-Ade) synthetic defined medium supplemented with X-α-Gal. Yeast harboring pGBKT7-53 and pGADT7-T vectors served as a positive control; yeast harboring pGBKT7-Lam and pGADT7-T vectors served as a negative control.

**Figure 8 ijms-23-12568-f008:**
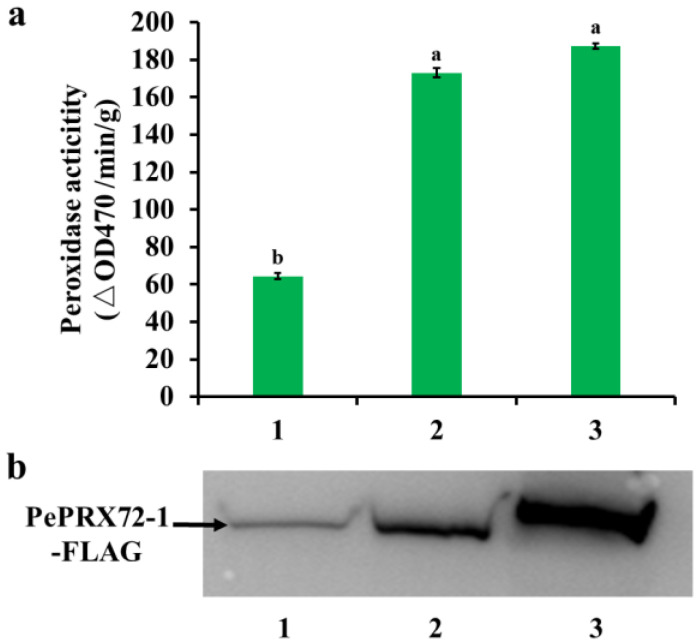
PeKFB9 attenuated the stability of PeRX72-1. (**a**) Peroxidase activity in crude extracts from tobacco leaves co-expressing PePRX72-1 with PeKFB9-FLAG (1), F^PeKFB9-FLAG (2), and pCAMBIA1300-3×FLAG (3). Data are the mean ±SD from three biological replicates. Different letters indicate significant differences (*p* < 0.01). (**b**) Immunoblot detection of the stability of the PePRX72-1-FLAG fusion proteins using anti-FLAG antibody in tobacco leaves co-infiltrated with PeKFB9-FLAG (1), F^PeKFB9-FLAG (2), and pCAMBIA1300-3×FLAG (3) constructs.

## Data Availability

All data generated or analyzed during this study are included in the article and its information files.

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
