# Peer review of "Identification of KFB Family in Moso Bamboo Reveals the Potential Function of PeKFB9 Involved in Stress Response and Lignin Polymerization"

_ijms, 2022, doi:10.3390/ijms232012568_

Round 1
Reviewer 1 Report
This study identified 19 KFBs in bamboo and analyzed their expression levels in different tissues or under abiotic stresses. In addition, they also found 2 proteins are physically interacted with PeKFB9. All the information generated by this study indeed laid a foundation for further improvement bamboo in future.
Minor spell error:
Line 11: "have" should be replaced by "has";
Line 30: "plan" should be "play"
Line 138: "is" should be "are"
Line 202: "were" change to "was"
Line 211: replace "and" with ","
Line 371 and Line 374: AtKFB01 and ATKFB1 should keep consist.
Line 423: "were" should be "was".
Two more questions"
1. Why you only did QPCR for 15 of 19 PeKFBs? you should clarify in the manuscript.
2. For Yeast two-hybrid screening, 16 clones were identified, only two genes (3 clones) were further investigated, what other clones are?
Author Response
Point 1: Line 11: "have" should be replaced by "has";
Response 1: Thank you for the suggestion. We have replaced "have" to "has".
Point 2: Line 30: "plan" should be "play"
Response 2: Thank you for the suggestion. We have replaced "plan" to "plays".
Point 3: Line 138: "is" should be "are"
Response 3: Thank you for the suggestion. We have replaced "is" to "are".
Point 4: Line 202: "were" change to "was"
Response 4: Thank you for the suggestion. We have replaced "were" to "was".
Point 5: Line 211: replace "and" with ","
Response 5: Thank you for the suggestion. We have replaced "and" to ",".
Point 6: Line 371 and Line 374: AtKFB01 and AtKFB1 should keep consist.
Response 6: Thank you for the suggestion. We have replaced "AtKFB1" to "AtKFB01" in Line 371.
Point 7: Line 423: "were" should be "was".
Response 7: Thank you for the suggestion. We have replaced "were" to "was".
Point 8: Why you only did QPCR for 15 of 19 PeKFBs? you should clarify in the manuscript.
Response 8: Suitable specific primers could not be designed for other PeKFBs. Transcriptome data generated from different tissue samples showed that the expression levels of these 4 PeKFBs were very low or even not detected in all the tissues, which may be the reason for the failure of suitable specific primers design. In addition, we have clarified in the manuscript.
Point 9: For Yeast two-hybrid screening, 16 clones were identified, only two genes (3 clones) were further investigated, what other clones are?
Response 9: Other clones are hypothetical protein, delta(8)-fatty-acid desaturase, unknown protein and so on. These clones are poor relevance to our study and will be studied in follow-up studies. The interaction protein of PeKFB9 from the library screening are shown in Table S3.

Reviewer 2 Report
In this paper, Yang et al, systematically identified 19 KFB proteins in moso bamboo and preliminarily explored the function of PeKFB9gene in stress response and lignin polymerization.The experimental work has been clearly presented and can basically explain the hypothesis proposed by the authors, which provides new insights of KFB proteins in moso bamboo. But, I have some minor points about the article that need to address.
Minor points:
1. The grammar of written English needs to be improved. For example, on line 30 “plan a central role” should be “plans a central role”. On line 68, “key regulator” should be “key regulators”. On line 323, “their function” should be “their functions”. The authors should carefully go through the whole manuscript to rectify any grammatical mistakes.
2. The quality of Figure 3a needs to be improved. The characters are blurred and unreadable.
3. Line 292. “Y2H assay results showed that experimental group didn't turn to blue.” It is not accurate that the authors use “not turn to blue” to describe non-interaction. I think “not turn to blue” could mean that they don't interact, or the non-blue yeast colonies is present.
Author Response
Point 1: The grammar of written English needs to be improved. For example, on line 30 “plan a central role” should be “plans a central role”. On line 68, “key regulator” should be “key regulators”. On line 323, “their function” should be “their functions”. The authors should carefully go through the whole manuscript to rectify any grammatical mistakes.
Response 1: Thank you for the suggestion. We have replaced "plan", "key regulator" and "their function" to "plays", "key regulators"and " their functions". In additions, we have carefully go through the whole manuscript.
Point 2: The quality of Figure 3a needs to be improved. The characters are blurred and unreadable.
Response 2: Thank you for the suggestion. We have improved the quality of Figure 3a. Please see the details in the revised manuscript.
Point 3: Line 292. “Y2H assay results showed that experimental group didn't turn to blue.” It is not accurate that the authors use “not turn to blue” to describe non-interaction. I think “not turn to blue” could mean that they don't interact, or the non-blue yeast colonies is present.
Response 3: Thank you for the suggestion. We have replaced "not turn to blue " to "grow on SD/-Ade/-His/-Leu/-Trp medium containing X-α-Gal". Please see the details in the revised manuscript.
